# High-altitude is associated with better short-term survival in critically ill COVID-19 patients admitted to the ICU

**Katherine Simbaña-Rivera[1], Pablo R. Morocho Jaramillo[2], Javier V. Velastegui Silva[3], Lenin Gómez-Barreno[1], Ana B. Ventimilla Campoverde[3], Juan F. Novillo Cevallos[2], Washington E. Almache Guanoquiza[3], Silvio L. Cedeño Guevara[2], Luis G. Imba Castro[2], Nelson A. Moran Puerta[3], Alex W. Guayta Valladares[2], Alex Lister[4], Esteban Ortiz-Prado[1,5]***

**1** One Health Research Group, Faculty of Health Science, Universidad de Las Americas, Quito, Ecuador, **2** Hospital Los Ceibos, Unidad de Terapia Intensiva, Guayaquil, Ecuador, **3** Hospital Quito Sur, Unidad de Terapia Intensiva, Quito, Ecuador, **4** University Hospital Southampton NHS FT, Southampton, United Kingdom, **5** Biomedicine Program, Department of Cell Biology, Physiology and Immunology Universidad de Barcelona, Barcelona, Spain

\* e.ortizprado@gmail.com

**Data Availability Statement:** All non-identifiable and previously anonymized data can be retrieved from the following link to our open data digital

## Abstract

### Background

Multiple studies have attempted to elucidate the relationship between chronic hypoxia and SARS-CoV-2 infection. It seems that high-altitude is associated with lower COVID-19 related mortality and incidence rates; nevertheless, all the data came from observational studies, being this the first one looking into prospectively collected clinical data from severely ill patients residing at two significantly different altitudes.

### Methods

A prospective cohort, a two-center study among COVID-19 confirmed adult patients admitted to a low (sea level) and high-altitude (2,850 m) ICU unit in Ecuador was conducted. Two hundred and thirty confirmed patients were enrolled from March 15[th] to July 15[th], 2020.

### Results

From 230 patients, 149 were men (64.8%) and 81 women (35.2%). The median age of all the patients was 60 years, and at least 105 (45.7%) of patients had at least one underlying comorbidity, including hypertension (33.5%), diabetes (16.5%), and chronic kidney failure (5.7%). The APACHE II scale (Score that estimates ICU mortality) at 72 hours was especially higher in the low altitude group with a median of 18 points (IQR: 9.5–24.0), compared to 9 points (IQR: 5.0–22.0) obtained in the high-altitude group. There is evidence of a difference in survival in favor of the high-altitude group (p = 0.006), the median survival being 39 days, compared to 21 days in the low altitude group.

repository: https://github.com/covid19ec/
HospitalData. Any additional query or information
about our research work can be requested to our
email address at e.ortizprado@gmail.com

**Funding:** The author(s) received no specific
funding for this work.

**Competing interests:** The authors have declared
that no competing interests exist.

**Abbreviations:** ACE2, Type-2 angiotensin-
converting enzyme; ARDS, Acute Respiratory
Distress Syndrome; BMI, Body Mass Index; CHF,
Congestive Heart Failure; CKD, Chronic Kidney
Disease; COPD, Chronic Obstructive Pulmonary
Disease; COVID-19, Novel Coronavirus disease
2019; CT, Computed Tomography; CVD,
Cerebrovascular Disease; ICU, Intensive care unit;
IQR, Interquartile Range; TB, Tuberculosis.

## Conclusion

There has been a substantial improvement in survival amongst people admitted to the high-
altitude ICU. Residing at high-altitudes was associated with improved survival, especially
among patients with no comorbidities. COVID-19 patients admitted to the high-altitude ICU
unit have improved severity-of-disease classification system scores at 72 hours.

## Introduction

In December 2019, the first cases of pneumonia due to the SARS-CoV-2 virus were reported
in Wuhan, China [1, 2]. On March 11[th], 2020, the novel Coronavirus disease (COVID-19), a
condition with multiple clinical features, which can rapidly evolve into acute respiratory dis-
tress syndrome (ARDS) and other serious complications, was declared a pandemic [2, 3]. The
disease spread rapidly, affecting regions and areas located in urban settings but also in rural
and geographically distant areas all over the world [4, 5].

The epidemiological behavior of the pandemic showed exponential growth throughout
many countries, while others seem to have managed the outbreak better [6]. Several studies
have identified differences in morbidity and mortality depending on many factors, including
socioeconomic status, the burden of chronic diseases, adequate access to health care, the
strength of the epidemiological surveillance systems, and the implementation of control mea-
sures as well as strict mobility restrictions [7]. During the first months of the pandemic, very
few ecological studies showed a possible epidemiological and survival implication exerted by
high-altitude [7, 8]. It has been proposed that these results are in part answered by the well-
known physiological acclimatization and the long-term adaptation to high-altitude exposure,
generating a greater tolerance to chronic hypoxia [7–10]. Several investigations have tried to
determine the potential relationship between high-altitude and COVID-19 related mortality
[8, 11–13]. Most studies have established that living at high-altitudes could be related to
reduced COVID-19 related mortality and morbidity [8, 11, 14, 15]. These results were dis-
cussed from different points of view. The biological one was based on a hypothesized lower
viral affinity for the type-2 angiotensin-converting enzyme (ACE2) receptors, but there is no
definitive evidence to support this claim [16]. Another hypothesis surrounding the high-alti-
tude-COVID-19 link refers to the involvement of better perfused and better-oxygenated tis-
sues due to the involvement of the hypoxia-triggered protein that regulates angiogenesis, cell
proliferation, metabolism, and down regulates ACE-2 levels; the well-known Hypoxia-induc-
ible factor 1-alpha (HIF-1 $\alpha$) [10, 17–22]. Having an improved ability to use oxygen within the
tissues might reduce the effects of systemic hypoxia caused by acute respiratory distress syn-
drome (ARDS) [23].

Contrariwise, socio-demographic and environmental factors such as population density,
UV radiation, ozone, or cold have been proposed to affect SARS-CoV-2 transmission and viral
load; nevertheless, no clinical data is available yet [24, 25].

The link between high-altitude hypoxia and COVID-19 mortality is still under investigation
[26, 27]. The very few studies on clinical, ventilatory, and respiratory support parameters' dif-
ferences have been performed at an elevation below 1,500 m, and no evidence about the role of
high-altitude exposure (> 2,500 m) on severally ill COVID-19 patients have been published
yet [23].

We believe our study is the first one that has been able to demonstrate the effect of altitude
living on COVID-19 mortality and prognosis after controlling for several clinical factors.

## Methods

### Study design

A prospective cohort study including patients with severe SARS-CoV-2 infection confirmed with real-time polymerase chain reaction (RT-PCR) was performed from March 15[th], 2020, to July 15[th,] 2020.

### Setting

The study was carried out in two intensive care units (ICU) located at two different elevations from the same Social Security Health System (IESS) in Ecuador. The ICU from the IESS-Quito Sur' Hospital located in the city of Quito (located at 2,850 m above sea level) and the IESS-Los Ceibos, located in the city of Guayaquil (Located at sea level).

Quito and Guayaquil are the most important cities in Ecuador, with 2.5 and 2.9 million inhabitants respectively; both have their IESS hospital built in 2017. They also share the distinction of becoming the first COVID-19 sentinel hospitals enabled in Ecuador. Since both hospitals are part of the same IESS social Security Health System, both were the first in receiving COVID-19 patients during the pandemic, and both are ruled by the same therapeutic guidelines; they are a good case-control opportunity to explore differences related to altitude.

### Population and study size

All the patients included in this study were admitted to the ICU unit in one of the two hospitals. The present study included a total of 230 patients diagnosed with COVID-19 using the RT-PCR technique, of which 114 patients were treated in the high-altitude group (IESS-Quito Sur), while 116 patients belonged to the low altitude group (IESS-Los Ceibos).

### Inclusion criteria

Adult men and women patients admitted to the ICU diagnosed with COVID-19 by means of RT-PCR who lived at least one year in the unit's coverage region and signed the informed consent for the use of the information was included in this study.

### Exclusion criteria

Patients diagnosed with COVID-19 by RT-PCR who did not meet the criteria for admission to the ICU or who lived less than one year in the unit's coverage region, or who did not sign the informed consent for the use of the information were excluded.

### Data sources and variables

Demographic information, clinical characteristics (including medical history, history of symptoms, comorbidities), chest computed tomography (CT) results, laboratory findings, ventilatory values, and medications used were collected from each patient. The dates of disease onset, hospital admission, admission to the ICU, and death or discharge date from the ICU were also recorded, as well as the APACHE II scores (Score that measures disease severity based on physiologic parameters, age, and previous health conditions) and the Charlson index (predicts 10-year survival in patients with multiple comorbidities). The onset date was defined as the day the patient noticed any symptoms. The severity of COVID-19 was defined according to the diagnosis and treatment guide for SARS-CoV-2 issued by the World Health Organization (WHO) [28]. It was designated as a critical illness due to COVID-19 when patients had one of the following criteria: (a) acute respiratory distress syndrome (any grade); (b) Septicemia; and

(c) septic shock. The data were obtained from the electronic medical record of a common registry system for both units and analyzed by three independent researchers.

## Statistical analysis

Categorical variables were summarized as frequencies and percentages, and continuous variables were described using median values and interquartile ranges (IQR) or mean and standard deviation, as appropriate. The analysis included a two-tailed Student's t-test, and the Mann-Whitney U test was used. The frequencies of the categorical variables were compared using the chi-square test and expressed in count and percentage. Also, survival curves (Kaplan Meier), the log-rank statistic, and the hazard ratio between groups were obtained.

Bivariate and multivariate analyzes were performed to identify factors associated with death from COVID-19 in all patients using the Cox risk regression model. To obtain a reduced set of variables from the broad set of predictors, we carried out a progressive *in bloc* procedure assigning the predictor variables into six groups: sociodemographic characteristics and comorbidities, complications, scales, ventilatory values, medications, and laboratory and imaging parameters. A multivariate regression analysis was applied within each block using two criteria to achieve the best set of predictors: relevance to the clinical situation and bivariate as well as multivariate statistical significance ($p < 0.05$) correcting for age. Variables with more than 25% missing values were not considered for the analysis.

All statistical analyzes were performed in SPSS version 25, and graphs were generated using GraphPad Prism version 7.00 software (GraphPad Software Inc).

## Bias

To minimize observation bias for systematic differences between the low and high-altitude group, observers who recollected data were blinded for the investigated hypothesis. To reduce investigation bias, coding and analysis were performed by three members of the research team independently, while discrepancies were resolved after achieving consensus.

## Ethical approval

This work was approved by the Hospital IESS-Quito-Sur Internal Review Board (IRB). The request for authorization was submitted on March 1st, 2020, and received ethical approval with the following identification number: ID: IESS-HG-SQ-CIE-2020-2656-M.

According to good clinical practices and local regulations, identifiable data from clinical records was only accessed by the medical team that was providing treatment and care to the patients.

# Results

The present study included a total of n = 230 patients diagnosed with COVID-19, of which n = 114 patients were treated in the high-altitude group, while 116 patients belonged to the low altitude group.

## Socio-demographic characteristic

The median age of all the patients was 60 years, with a range of 49 to 69 years, and the majority (80.9%) of them were over 45 years of age. More than half (64.8%) of the patients were men. The BMI median of all the patients was 27.8 kg/m$^2$, while about half (47.8%) were overweight, and (32.9%) some degree of obesity. A total of n = 105 (45.7%) patients had at least one underlying comorbidity, the most frequent being chronic diseases, such as hypertension (33.5%), diabetes (16.5%), and chronic kidney failure (5.7%). Five patients with COPD were identified

**Table 1. Analysis of the chi-square, mean and median differences for demographic and independent risk factors in COVID-19 critically ill patients living at low and high-altitudes which were hospitalized in intensive care units.**

| Category | All | High-altitude | Low altitude | P-value |
|---|---|---|---|---|
| | n (%) | n (%) | n (%) | |
| Age—median (IQR) | 60.0 (49.0–69.0) | 55.5 (49.0–66.0) | 62.5 (48.5–69.0) | 0.181 |
| Sex | | | | |
| Male | 149 (64.8) | 77 (67.5) | 72 (62.1) | 0.385 |
| Female | 81 (35.2) | 37 (32.5) | 44 (37.9) | 0.385 |
| BMI—median (IQR) | 27.8 (25.7–30.9) | 27.4 (24.9–31.0) | 28.2 (26.0–30.7) | 0.285 |
| Underweight | 1 (0.4) | 1 (0.9) | 0 (0.0) | 0.096 |
| Normal weight | 43 (18.9) | 27 (24.1) | 16 (13.8) | 0.096 |
| Overweight | 109 (47.8) | 48 (42.9) | 61 (52.6) | 0.096 |
| Obesity class I | 59 (25.9) | 25 (22.3) | 34 (29.3) | 0.096 |
| Obesity class II | 15 (6.6) | 10 (8.9) | 5 (4.3) | 0.096 |
| Obesity class III | 1 (0.4) | 1 (0.9) | 0 (0.0) | 0.096 |
| Charlson index | | | | |
| Presence of Comorbidities | 135 (58.7) | 85 (74.6) | 50 (43.1) | 0.000 |
| Low comorbidity | 28 (12.2) | 10 (8.8) | 18 (15.5) | 0.000 |
| Hight comorbidity | 67 (29.1) | 19 (16.7) | 48 (41.4) | 0.000 |
| Comorbidity | 105 (45.7) | 33 (28.9) | 72 (62.1) | 0.000 |
| Arterial Hypertension | 77 (33.5) | 21 (18.4) | 56 (48.3) | 0.000 |
| CKD | 13 (5.7) | 3 (2.6) | 10 (8.6) | 0.049 |
| Asthma | 2 (0.9) | 1 (0.9) | 1 (0.9) | 0.99 |
| Diabetes | 38 (16.5) | 12 (10.5) | 26 (22.4) | 0.015 |
| Psoriasis | 2 (0.9) | 0 (0.0) | 2 (1.7) | 0.159 |
| CHF | 2 (0.9) | 1 (0.9) | 1 (0.9) | 0.99 |
| Cancer | 4 (1.7) | 2 (1.8) | 2 (1.7) | 0.986 |
| CVD | 2 (0.9) | 0 (0.0) | 2 (1.7) | 0.159 |
| COPD | 5 (2.2) | 4 (3.5) | 1 (0.9) | 0.169 |
| Rheumatoid arthritis | 2 (0.9) | 0 (0.0) | 2 (1.7) | 0.159 |
| TB | 2 (0.9) | 2 (1.8) | 0 (0.0) | 0.152 |
| Hepatic cirrhosis | 2 (0.9) | 1 (0.9) | 1 (0.9) | 0.99 |
| Pulmonary fibrosis | 3 (1.3) | 1 (0.9) | 2 (1.7) | 0.571 |
| Hypothyroidism | 5 (2.2) | 3 (2.6) | 2 (1.7) | 0.637 |

Abbreviations: IQR: Interquartile range; BMI: body mass index; CKD: Chronic kidney disease; CHF: Chronic hepatic failure; CVD: Cerebro-vascular Disease; COPD: Chronic obstructive pulmonary disease; TB: Tuberculosis.

(Table 1). When comparing the samples by altitude, no differences were evidenced between age, sex, and BMI. On the other hand, the low altitude group presented a greater number of patients with comorbidities measured by the Charlson index, highlighting the cases of hypertensive and diabetic patients. The mean interval from the onset of symptoms to admission to the ICU for all patients was eight days (IQR: 6–11). However, in the high-altitude group, there was a shorter median of 7 days (IQR: 5–10) (Table 1).

## Clinical characteristics

Regarding the scales evaluated, it was evidenced that upon admission to the ICU, the APACHE II scale in the first 24 hours presented a median of 15 points (IQR: 10.0–20.0) in the high-altitude group, whilst the low altitude scored 16 points (12.0–20.5) and did not show a statistical difference for both groups (p = 0.206). However, the same scale at 72 hours was especially

**Table 2. Analysis of the chi-square, mean and median differences for clinical predictors for COVID-19 mortality among critically ill patients living at low and high-altitudes which were hospitalized in intensive care units.**

| Category | Measure | All | High-altitude | Low altitude | P-value |
|---|---|---|---|---|---|
| Symptom's onset | median (IQR) | 8.0 (6.0–11.0) | 7.0 (5.0–10.0) | 8.0 (7.0–13.0) | 0.003 |
| Waiting time before admission in the UCI (H) | median (IQR) | 2.3 (0.0–8.2) | 3.0 (0.0–10.0) | 0.7 (0.0–8.0) | 0.01 |
| Condition of discharge from ICU | | | | | |
| Dead | n (%) | 129 (56.1) | 52 (45.6) | 77 (66.4) | 0.002 |
| Alive | n (%) | 101 (43.9) | 62 (54.4) | 39 (33.6) | 0.002 |
| ApacheII ICU (24H) | median (IQR) | 16.0 (11.0–20.0) | 15.0 (10.0–20.0) | 16.0 (12.0–20.5) | 0.206 |
| ApacheII ICU (72H) | median (IQR) | 14.0 (6.0–23.0) | 9.0 (5.0–22.0) | 18.0 (9.5–24.0) | 0.001 |
| Shock | | | | | |
| No shock | n (%) | 64 (27.8) | 34 (29.8) | 30 (25.9) | 0.503 |
| Septic shock | n (%) | 114 (49.6) | 41 (36.0) | 73 (62.9) | 0.503 |
| Distributive shock | n (%) | 46 (20.0) | 37 (32.5) | 9 (7.8) | 0.503 |
| Obstructive shock | n (%) | 3 (1.3) | 2 (1.8) | 1 (0.9) | 0.503 |
| Cardiogenic shock | n (%) | 3 (1.3) | 0 (0.0) | 3 (2.6) | 0.503 |
| Respiratory (ARDS) | n (%) | 219 (95.2) | 106 (93.0) | 113 (97.4) | 0.115 |
| Renal | | | | | |
| Did not present fault | n (%) | 129 (56.1) | 61 (53.5) | 68 (58.6) | 0.094 |
| Acute renal failure | n (%) | 89 (38.7) | 50 (43.9) | 39 (33.6) | 0.094 |
| Exacerbated chronic kidney failure | n (%) | 12 (5.2) | 3 (2.6) | 9 (7.8) | 0.094 |
| Dialysis | n (%) | 26 (11.3) | 11 (9.6) | 15 (12.9) | 0.431 |
| Coagulation | n (%) | 17 (7.4) | 6 (5.3) | 11 (9.5) | 0.221 |
| Polyneuropathy | n (%) | 85 (37.0) | 40 (35.1) | 45 (38.8) | 0.561 |
| Delirium | n (%) | 88 (38.3) | 45 (39.5) | 43 (37.1) | 0.708 |
| Hypoxic encephalopathy | n (%) | 3 (1.3) | 3 (2.6) | 0 (0.0) | 0.079 |
| Hepatic | n (%) | 30 (13.0) | 17 (14.9) | 13 (11.2) | 0.528 |

Abbreviations: H: Hours; ICU: Intensive care unit; ARDS: Acute respiratory distress syndrome.

higher in the low altitude group with a median of 18 points (IQR: 9.5–24.0), compared to 9 points (IQR: 5.0–22.0) obtained in the group of high-altitude. Concerning the most common complications presented during the ICU stay, acute respiratory distress syndrome in adults was evidenced in n = 219 (95.2%) patients, any type of shock in n = 166 (72.2%) patients, acute / exacerbated renal failure in n = 101 (43.9%) and delirium in n = 88 (38.3%). There were no statistical relationships between complications and altitude (Table 2). Finally, n = 129 deaths (56.1%) were recorded in the entire sample, of which most (n = 77 (66.4%) were recorded in the low altitude group compared to n = 52 (45.6%) at high-altitude, p = 0.002 (Table 2).

The laboratory results are shown in (Table 3), where lower values of platelets, liver enzymes (AST and ALT), and lactate were evidenced in the low altitude group compared to the high-altitude group. Against the leukocyte count was higher in the low altitude group (Fig 1).

The acid-base profile for both groups show normal ranges; however, there is a greater alkalotic component (higher median pH and bicarbonate) in the low altitude group (p = 0.000). At the same time, the level of $CO_2$ appears in normal ranges and without difference between the two groups (Table 3).

## Ventilatory findings

Ventilatory management in both groups was based on the critical care COVID-19 Treatment Guidelines approved by the National Institute of Health (NIH) and the current

**Table 3. Analysis of the mean and median differences of the principal hematological and serological parameters in severely ill patients with covid19.**

| Category | All | High-altitude | Low altitude | P-value |
|---|---|---|---|---|
| | median (IQR) | median (IQR) | median (IQR) | |
| **Hematic Biometry** | | | | |
| Hemoglobin mg/dL | 13.7 (12.2–14.7) | 13.7 (12.4–14.8) | 13.5 (11.8–14.6) | 0.095 |
| Leukocytes $10^3/\mu L$ | 11.8 (8.9–16.1) | 10.5 (8.2–14.4) | 13.0 (10.2–17.4) | 0.000 |
| Leukocytes $10^3/\mu L$ (7D) | 11.8 (9.3–16.9) | 10.8 (8.7–12.7) | 13.5 (10.2–20.7) | 0.000 |
| Neutrophils % | 86.7 (81.0–90.0) | 87.0 (81.0–90.0) | 86.2 (81.3–89.5) | 0.215 |
| Lymphocytes $10^3/\mu L$ | 6.7 (4.2–10.2) | 6.7 (4.4–10.0) | 6.1 (4.0–10.9) | 0.994 |
| Platelets $10^3/\mu L$ | 290.0 (215.0–378.0) | 277.0 (218.0–367.0) | 300.5 (213.0–387.0) | 0.605 |
| Platelets $10^3/\mu L$ (72H) | 283.0 (202.0–369.0) | 300.5 (228.0–383.0) | 255.5 (185.0–347.0) | 0.015 |
| Platelets $10^3/\mu L$ (7D) | 262.0 (185.0–401.0) | 306.0 (207.0–435.0) | 241.0 (152.0–338.0) | 0.002 |
| **Blood chemistry** | | | | |
| D-Dimer ng/ml | 2,193.0 (796.0–6,293.0) | 1,895.5 (743.0–4,745.0) | 2,600.0 (800.0–7,700.0) | 0.475 |
| D-Dimer ng/ml (72H) | 2,003.0 (738.0–5,400.0) | 2,347.0 (947.0–3,800.0) | 1,600.0 (450.0–6,700.0) | 0.745 |
| Urea mg/dL | 42.0 (27.0–64.0) | 36.4 (23.5–57.7) | 45.5 (30.0–66.5) | 0.011 |
| Creatinine mg/dL | 0.8 (0.7–1.2) | 0.8 (0.7–1.1) | 0.8 (0.7–1.3) | 0.150 |
| Ferritin $\mu g$/ml | 1,469.0 (912.2–2,000.0) | 1,418.8 (824.0–1,979.0) | 1,600.0 (1,047.0–2,000.0) | 0.300 |
| LDH U/L | 402.5 (315.0–584.0) | 382.0 (303.0–541.0) | 427.0 (320.0–608.0) | 0.134 |
| CRP mg/L | 27.0 (18.5–45.0) | 22.8 (18.1–33.3) | 36.0 (19.0–48.0) | 0.004 |
| PCT ng/ml | 0.5 (0.2–1.8) | 0.5 (0.2–1.9) | 0.6 (0.2–1.6) | 0.953 |
| **Gasometry** | | | | |
| ph | 7.4 (7.3–7.4) | 7.4 (7.3–7.5) | 7.3 (7.2–7.4) | 0.000 |
| ph (7D) | 7.4 (7.3–7.5) | 7.4 (7.3–7.5) | 7.4 (7.3–7.4) | 0.001 |
| $SaO_2$% | 94.0 (89.0–97.0) | 91.0 (87.0–94.0) | 96.9 (93.0–98.0) | 0.000 |
| $SaO_2$% (7D) | 96.0 (92.0–98.0) | 94.0 (91.0–96.0) | 97.7 (96.0–98.2) | 0.000 |
| $PaO_2$ mmHg | 76.0 (59.0–104.0) | 65.0 (52.0–76.0) | 97.9 (72.3–138.4) | 0.000 |
| $PaO_2$ mmHg (7D) | 82.5 (67.0–119.8) | 69.5 (61.3–80.7) | 110.5 (88.0–145.0) | 0.000 |
| $PaCO_2$ mmHg | 38.0 (31.7–48.3) | 35.8 (31.0–43.0) | 43.0 (32.5–56.3) | 0.007 |
| $PaCO_2$ mmHg (7D) | 42.0 (38.0–51.0) | 42.7 (38.0–52.0) | 41.9 (37.2–50.2) | 0.774 |
| $HCO_3$ mEq/L | 20.9 (18.7–23.6) | 21.9 (19.1–24.9) | 20.2 (17.7–22.6) | 0.000 |
| $HCO_3$ mEq/L (7D) | 25.3 (21.5–29.1) | 28.0 (22.1–30.6) | 24.2 (20.0–26.9) | 0.000 |
| Lactate mmol/L | 1.8 (1.3–2.3) | 1.8 (1.4–2.3) | 1.7 (1.0–2.1) | 0.800 |

Abbreviations: H: Hours; D: Days; ALT: Alanine aminotransferase; AST: Aspartate aminotransferase; LDH: lactate dehydrogenase; PCR: C-reactive protein; PCT: procalcitonin; ph: Potential of hydrogen; SaO2: Oxygen saturation of arterial blood; PaO2: Partial pressure of oxygen in arterial blood; PaCO2: Partial pressure of carbon dioxide in arterial blood; HCO3: Serum bicarbonate.

recommendations of the emergency committee for the management of COVID-19 in Ecuador (COE). In that sense, there were very few differences in the parameters used to ventilate patients in the low or high-altitude group. The tidal volume differed in less than 7cmH$_2$O, and either the peak or the pulmonary compliance varied significantly based on several guidelines [29–32]. Although there was a difference in terms of the PEEP pressure, this is often seen when using high pressures during ventilation [33, 34].

Mechanical ventilation was maintained for a median of 12 days (IQR: 7.0–20.0) for the two groups. During admission, the FiO$_2$ for both groups did not show differences; however, the measurements at 72 hours and seven days show significant differences with higher values in the low altitude group (Fig 2).

Similarly, a higher value was evidenced during the measurements of the partial pressure of oxygen (PO$_2$) in the patients in the low altitude group, compared to the high-altitude group

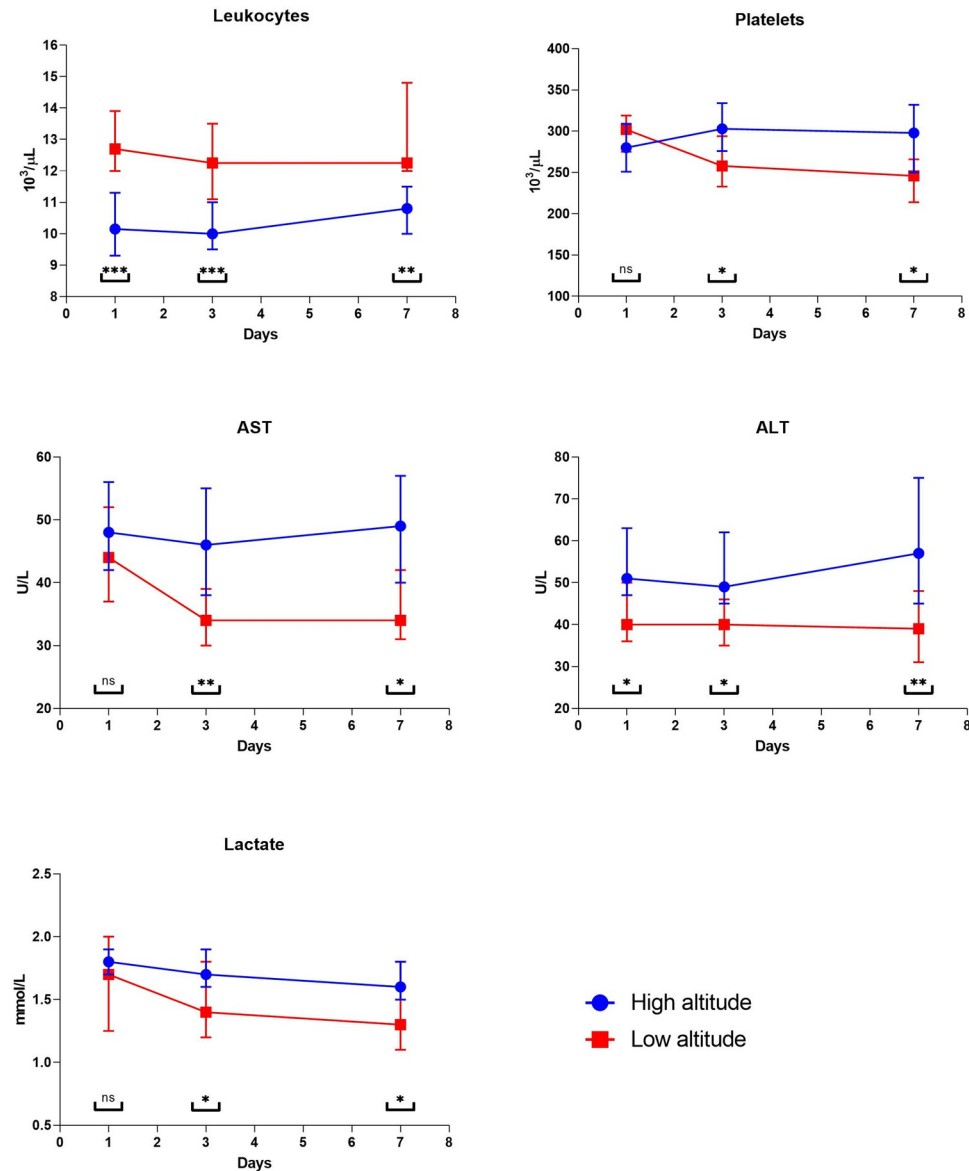

**Fig 1. Statistical hematological differences between the low and high-altitude group.** All comparisons were calculated using a U Mann-Whitney test * = <0.05; ** = <0.01;*** = <0.001; ns = non-significant. Abbreviations: ALT: Alanine aminotransferase; AST: Aspartate aminotransferase.

and consequently in the Pa-Fi relationship (Fig 2). The need for a tracheostomy reached 18.7% of patients (Table 4).

## Medicines

During the hospital stay, 136 (63.6%) patients received corticosteroids, of which up to 60 (39%) in mg/kg doses, for a median time of 4 (IQR: 3–6) days. The most prescribed corticosteroid was methylprednisolone (n = 97; 42.2%). A total of n = 224 (97.4%) of patients received heparins, of which 82 (79.1%) patients received isocoagulation doses. Regarding medications that to date were used for the treatment of COVID, it was evidenced that 118 (51.3%) and 149 (64.8%) of patients received Hydroxychloroquine and Lopinavir / Ritonavir, respectively. In

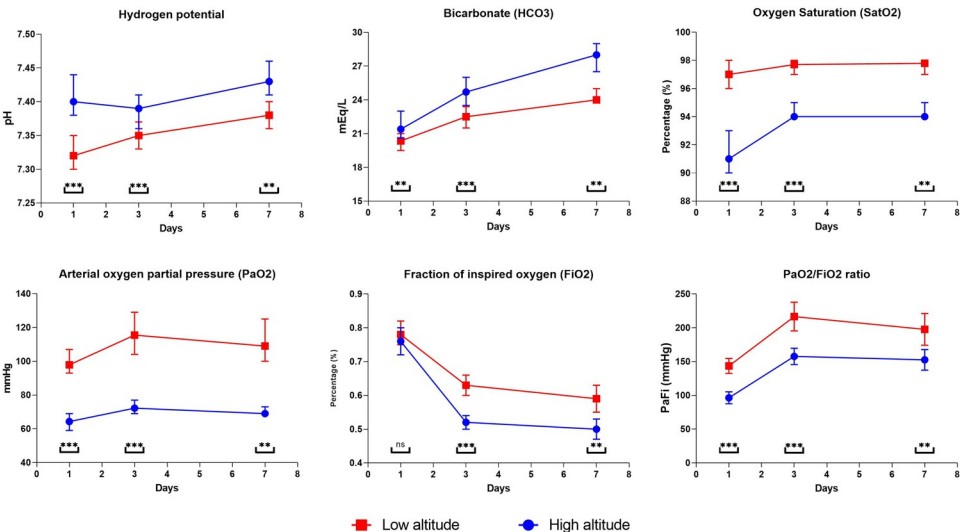

**Fig 2. Statistical differences among respiratory and blood serum parameters among COVID-19 patients living at two different elevations.** All comparisons were calculated using a U Mann-Whitney. Except for bicarbonate on seventh day * = <0.05; ** = <0.01;*** = <0.001; ns = non-significant.

the comparison between groups, the low altitude group received greater numbers of corticosteroid prescriptions and tocilizumab than the high-altitude group. However, the patients did not show differences in the administration of heparins, antimalarials, or lopinavir/ritonavir (Table 5).

**Table 4. Ventilatory and pulmonary parameters among COVID-19 patients in the low and high-altitude group.**

| Category | Measure | All | High-altitude | Low altitude | P-value |
|---|---|---|---|---|---|
| Received mechanical ventilation | n (%) | 204 (88.7) | 105 (92.1) | 99 (85.3) | 0.105 |
| High Flow | n (%) | 12 (5.2) | 0 (0.0) | 12 (10.3) | 0.000 |
| Recruitment | n (%) | 72 (31.3) | 33 (28.9) | 39 (33.6) | 0.445 |
| Tracheostomy | n (%) | 43 (18.7) | 23 (20.2) | 20 (17.2) | 0.568 |
| Pronation | | | | | |
| Intermittent | n (%) | 56 (24.3) | 18 (15.8) | 38 (32.8) | 0.014 |
| Extended | n (%) | 126 (54.8) | 68 (59.6) | 58 (50.0) | 0.014 |
| Pronation (H) | median (IQR) | 50.0 (16.0–96.0) | 48.0 (2.0–72.0) | 72.0 (24.0–120.0) | 0.000 |
| PAFI mmHg | median (IQR) | 104.3 (72.9–154.5) | 87.0 (61.0–121.0) | 133.1 (94.4–176.6) | 0.000 |
| PAFI mmHg (72H) | median (IQR) | 165.0 (128.3–216.7) | 150.0 (125.5–192.4) | 187.5 (147.2–255.0) | 0.000 |
| PEEP (cmH2O) | median (IQR) | 12.0 (10.0–14.0) | 12.0 (10.0–14.0) | 10.0 (9.0–12.0) | 0.000 |
| PEEP (cmH2O) (72H) | median (IQR) | 10.0 (9.0–12.0) | 12.0 (9.0–14.0) | 10.0 (8.0–12.0) | 0.002 |
| Peak Pressure | median (IQR) | 30.0 (28.0–32.0) | 28.0 (25.0–31.0) | 31.0 (29.0–35.0) | 0.000 |
| Peak Pressure (72H) | median (IQR) | 30.0 (26.0–32.0) | 28.0 (24.0–30.0) | 31.0 (28.0–33.0) | 0.000 |
| Plateau Pressure (cmH2O) | median (IQR) | 26.0 (23.0–28.0) | 25.0 (22.0–28.0) | 26.0 (24.0–29.0) | 0.047 |
| Plateau Pressure (cmH2O) (72H) | median (IQR) | 25.0 (21.0–27.0) | 24.0 (20.0–27.0) | 25.5 (22.0–28.0) | 0.029 |
| Static Compliance (ml/cmH20) | median (IQR) | 28.0 (22.0–37.0) | 30.5 (26.0–37.0) | 27.0 (18.0–34.0) | 0.006 |
| Static Compliance (ml/cmH20) (72H) | mean (SD) | 31.65 (10.92) | 33.63 (9.48) | 29.53 (11.97) | 0.008 |
| Driving pressure (cmH2O) (24H) | median (IQR) | 13.0 (9.0–16.0) | 12.0 (9.0–15.0) | 14.0 (8.0–18.0) | 0.022 |
| Driving pressure (cmH2O) (72H) | median (IQR) | 12.5 (9.0–16.0) | 12.0 (8.0–14.0) | 14.0 (9.0–17.0) | 0.021 |

Abbreviations: H: Hours; D: Days; FiO2: Fraction of inspired oxygen; RR: Respiratory rate; PAFI: Relationship between the alveolar-arterial oxygen gradient and PaO2/FiO2; PEEP: Positive End-Expiratory Pressure.

**Table 5. Pharmaceutical treatment in the low and high-altitude group.**

| Category | All | High-altitude | Low altitude | P-value |
|---|---|---|---|---|
| Heparins | 224 (97.4) | 71 (98.3) | 112 (96.6) | 0.420 |
| Heparin at isocoagulation doses | 182 (79.1) | 71 (62.3) | 111 (95.7) | 0.000 |
| Heparin at anticoagulation doses | 42 (18.3) | 41 (36.0) | 1 (0.9) | 0.000 |
| Corticosteroids | 146 (63.6) | 59 (51.8) | 87 (75.0) | 0.000 |
| Methylprednisolone | 97 (42.2) | 40 (35.1) | 57 (49.1) | 0.000 |
| Dexamethasone | 34 (14.8) | 5 (4.4) | 29 (25.0) | 0.000 |
| Hydrocortisone | 2 (0.9) | 2 (1.8) | 0 (0.0) | 0.000 |
| Prednisone | 13 (5.7) | 12 (10.5) | 1 (0.9) | 0.000 |
| Corticosteroids Days—median (IQR) | 4.0 (3.0–6.0) | 3.0 (3.0–3.0) | 5.0 (3.0–7.0) | 0.000 |
| Corticosteroids Doses | | | | |
| mg/kg | 60 (39.0) | 26 (43.3) | 34 (36.2) | 0.649 |
| Lower Doses | 48 (31.2) | 18 (30.0) | 30 (31.9) | 0.649 |
| Pulses | 46 (29.9) | 16 (26.7) | 30 (31.9) | 0.649 |
| Antimalarials | 168 (73.0) | 88 (77.2) | 80 (69.0) | 0.312 |
| Hydroxychloroquine | 50 (21.7) | 13 (11.4) | 37 (31.9) | 0.000 |
| Chloroquine | 118 (51.3) | 75 (65.8) | 43 (37.1) | 0.000 |
| Others | | | | |
| Lopinavir/Ritonavir | 149 (64.8) | 69 (60.5) | 80 (69.0) | 0.180 |
| Tocilizumab | 23 (10.0) | 3 (2.6) | 20 (17.2) | 0.000 |

## Altitude and mortality from COVID

There is evidence of a difference in survival in favor of the high-altitude group (p = 0.006), with the median survival of 39 days, compared to 21 days of the low altitude group (Fig 3).

The hazard ratio obtained was 0.55 (95%CI = 0.39–0.78). Due to the differences found regarding comorbidities and age, the subgroups analysis was performed based on the Charlson classification (Fig 4).

It was evident that the patients classified with a low and high Charlson index did not present differences between groups by altitude p = 0.929 and p = 0.920, respectively. However, in the group with no comorbidities, there was evidence of a difference between altitudes (p = 0.005), with the median survival of 17 days in the low altitude group and 49 in the high-altitude group. The hazard ratio found was 0.41 (95% CI = 0.23–0.75).

In the age subgroup analysis, it was demonstrated that the patients with an age between 51–65 years and over 65 years did not present differences by altitude p = 0.250 and p = 0.097, respectively. In contrast, within the younger group (those under 50 years), we found statistically significant differences between altitudes (p = 0.001), with the median survival of 19 days in the low altitude group and 66 in the high-altitude group. The hazard ratio found was 0.27 (95% CI = 0.11–0.64) (Fig 5).

## Predictors of death

In the final adjusted analysis, five factors associated with the risk of death were found. As a protection factor, they were high-altitude and the presence of a tracheostomy. On the other hand, as risk factors were the APACHE II score greater than 17 at 72 hours, the relationship between arterial oxygen pressure and inspiratory oxygen fraction ($PaO_2 / FiO_2$) on the seventh day less than 300, and the presence of coagulopathy during the hospitalization (Fig 6).

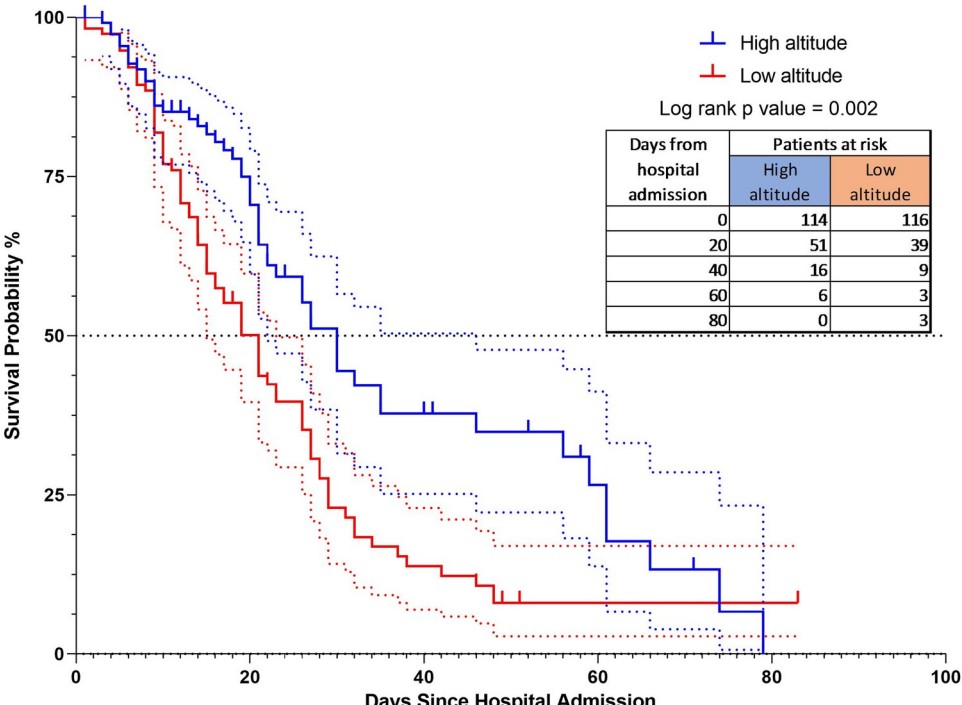

**Fig 3. The Kaplan-Meier curves for mortality according to altitude.** All comparisons were calculated using a COX regression multivariate analysis.

## Discussion

The COVID-19 pandemic has severely affected the normal functioning of all countries on the planet, with a greater impact on developing countries. The impact of the pandemic caused by the COVID-19 pandemic has been more striking in countries with weakened health systems, causing thousands of deaths attributed to SARS-CoV-2 infections [35–38]. Although several factors have been linked to lower or higher COVID-19 related attack or mortality rates, other factors such as hypobaric hypoxia found at high-altitudes have been proposed as possible covariates. Evidence has been found around a reduced attack rate due to SARS-CoV-2 infection and lower mortality rates in those places located at higher altitudes [7, 15, 25, 39]. The pathophysiological reason for this relationship has not been established yet. Nevertheless, several hypotheses have been proposed. For instance, it seems like the current evidence suggests that lower COVID-19 related deaths are attributed either to a biological adaptation to hypoxia among high-altitude dwellers, due to environmental factors such as high UV or ozone exposure, or more logical; due to demographic denistity [25]. High-altitude patients are exposed to hypoxia, increasing molecular levels of HIF-1α and HIF-2α, which might favors a greater tolerance to hypoxemia and decreases the acute tissue damage triggered by patients with severe acute respiratory conditions [40]. Nevertheless, a recent study denies this [41]. Tian et al., 2021 concluded that SARS-CoV-2 ORF3a and host hypoxia-inducible factor-1α (HIF-1α) play key roles in the virus infection and pro-inflammatory responses, dysregulating oxygen metabolism among COVID-19 patients [41]. On the other hand, it has been shown that some high-altitude resident population groups have developed polymorphisms of the ACE-2 receptor that favor a better tolerance to hypoxia [42, 43]. The ACE-2 receptors, when inactivated, promote a pro-inflammatory state that would increase the repercussions in the lungs and other organs [43–45]. Other physiological mechanisms could justify, at least in part, this apparent protection

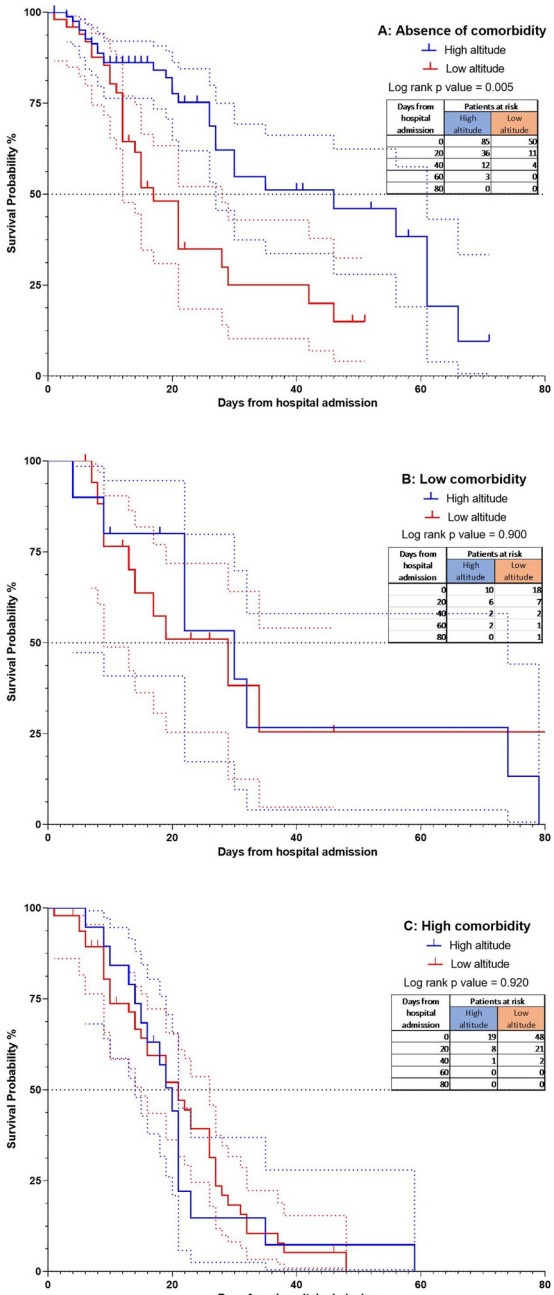

**Fig 4. The Kaplan-Meier curves for mortality according to altitude and group analysis using the Charlson index.**
4A Curve with absence of comorbidities p = 0.005; 4B Curve with low comorbidities p = 0.900; 4C Curve with high
comorbidities p = 0.920. All comparisons were calculated with COX regression multivariate analysis.

conferred by geographical altitude. It is believed that at high-altitudes there is a lower expres-
sion of ACE-2 receptors, which are precisely the gateway to our cells for the SARS-CoV-2
virus [8]. A more plausible explanation goes along with the fact that high-altitude inhabitants
express genes responsible for producing more erythrocytes (increasing oxygen transport) and
creating new blood vessels (greater oxygen supply) [10, 46, 47]. On top of this, we must add
certain anatomical and morphological characteristics among high-altitude dwellers, such as

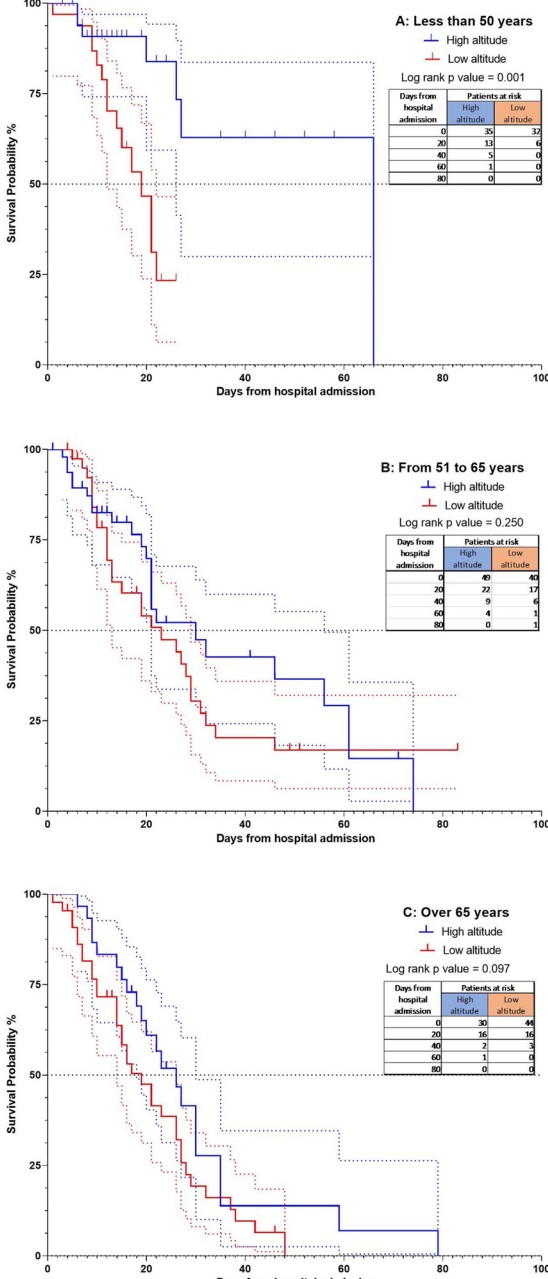

**Fig 5. The Kaplan-Meier curves for mortality according to altitude and age categories.** 5A Curve with age less than 50 years p = 0.001; 5B Curve with age from 51 to 65 years p = 0.250; 5C Curve with age over 65 years p = 0.097. All comparisons were calculated with COX regression multivariate analysis.

larger and bigger thoraxes as well as greater ventilatory capacities, that might play a role in reducing hypoxia found during severe ARDS due to COVID-19 [23, 48–50].

Despite the absence of clear pathophysiology, the present study provides relevant epidemiological and clinical data for understanding the influence of altitude on the evolution of seriously ill patients with COVID-19. Our results show a male to female predominance in terms of hospital admission, having a median age of 60 years, results comparable to other already published [3, 51–53]. The biochemical analysis from our cohorts showed the typical spectrum

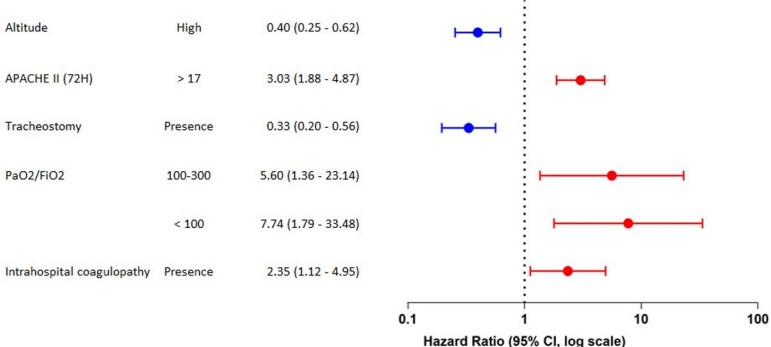

**Fig 6. High altitude mortality predictors among COVID-19 patients living at two different altitudes.**

of hematological alterations, including leukocytosis with neutrophilia and lymphopenia, increased values of plasma ferritin, LDH, CRP, and abnormal platelet count and procalcitonin levels as evidenced in most patients elsewhere [54–58].

The high-altitude group had a higher number of leukocytes, a lower number of platelets, and higher levels of CRP; however, the rest of the values were similar in both groups. For the oxygenation parameters expressed in levels of $PaO_2$ / $FiO_2$, $PO_2$, and $SaO_2$ we found that in both groups, a hypoxic profile was evident at the time of admission. These findings are often present in patients with dyspnea, increased heart rate, and decreased $PaO_2$ / $FiO_2$ value, a common scenario for COVID-19 patients admitted to the ICU [59]. In terms of ventilatory parameters, the median static pulmonary compliance ranged from 27 to 35 ml/cm $H_2O$ as reported elsewhere [60–62]. However, the high-altitude group showed lower oxygenation values with higher static compliance. This lack of correlation between $PaO_2$ / $FiO_2$ and static compliance in patients with COVID-19 was also reported by Grasseli et al., 2020 [60]. The explanation behind this finding could be linked to the act that COVID-19 lungs have vascular alterations secondary to endothelial damage [63].

Our results demonstrate that the presence of overweight and obesity were consistent characteristics of both groups, similar findings as reported previously [64]. Current evidence is clear linking obesity as an independent predictor of mortality among COVID-19 patients [65]. Our study has similar results to a large UK study, which confirmed that 44% of hospitalized patients were overweight and 34% obese [65]. The information suggests that after adjusting for possible confounding factors, including age, sex, ethnicity, and social deprivation, the relative risk of critical illness from COVID-19 increases by 44% for overweight people and almost doubles for obese people.

As observed, comorbidities are well-associated with an increasing risk of COVID-19 related mortality [66, 67]. We found that mortality is positively associated with having one or more comorbidities; however, in our study, we found that although the presence of comorbidities is higher in populations located at lower altitudes, once we excluded the presence of comorbidities from the model, the hypothesized protective effect of high-altitude is evident. In other words, patients with comorbidities are at higher risk of dying at both altitudes when compared to patients with no comorbidities, and nevertheless, when compared only patients without comorbidities from the low and high-altitude group, we found that highlanders have a greater chance of survival. In the same way, age is an important death risk factor for COVID-19 infection. In our study, we found that oldest people have higher death cases. But when survival analysis was stratified by age groups. It was evidence similar patterns in the oldest groups and surprisingly only under 50-year-old group had a statistical difference.

The treatment basis for moderate-severe ARDS secondary to COVID-19 is based on ventilatory support with low tidal volumes, a prone position, and active management of intravenous fluids [23, 28, 68, 69]. In this sense, it was evident that the strategy the ventilator used for both groups in the present study was protective and up to 79.1% of patients complied with the prone position. Despite these measures, the complications were not different, and the high-altitude group was characterized by higher survival with a median of 39 days, compared to 21 days in the low altitude group. A difference that remained constant in the subgroup without comorbidities provides one of the first measurements of the contribution of altitude as a factor to be considered in this pathology. It is also important to note that currently, the evidence for the use of dexamethasone has been established as a protective factor for mortality [70]; however, the low altitude group that received a greater amount of corticosteroids presented higher mortality than the high-altitude group, supporting, even more, our hypothesis that high-altitude residents tolerated systemic hypoxia better than their low land peers.

In the Cox regression model, five factors associated with the risk of death were evidenced. These findings of increased risk did not differ from what is reported in the literature [71]. In spite of this, the APACHE II score has been an effective clinical tool to predict hospital mortality in patients with COVID-19 [72]. Thromboembolic events have also been shown to pose a significant risk of mortality in critically ill patients [73]. Regarding the $SO_2 / FiO_2$ ratio, it has been widely studied as a prognostic factor [74]. Besides, as protection, tracheostomy is a procedure that favors the release of ventilatory support in patients with prolonged mechanical ventilation, for which it has been classified as a protective factor against severe complications [75].

Finally, although altitude has been reported as a possible intervening factor in the clinical outcome of several patients [7, 8], this is the first study to elucidate more causal information on this relationship.

## Limitations

This study was carried out in two COVID-19 designed hospitals, part of the social security health system (IESS). The private for-profit health system also receives COVID-19 patients; nevertheless, this population was not included in our analysis. An important limitation is a fact that we cannot adjust respiratory or oxygenation parameters for both populations, as the presence of validated data at these altitudes is scarce. Although our findings are the first to try to identify the role of altitude in relation to COVID-19 related mortality, further studies, including those with a broader altitude range, could provide important information to resolve some of the uncertainties surrounding our research question.

## Conclusion

In this series of cases of critically ill patients with COVID-19 who were admitted to the ICU, there has been a substantial improvement in survival amongst people admitted to the high-altitude critical care unit. High-altitude living was associated with improved survival, especially among patients with no comorbidities. COVID-19 patients admitted to the high-altitude ICU unit have improved severity-of-disease classification system scores at 72 hours and reported better respiratory and ventilatory profiles than the low altitude group. Our analysis suggests this improvement is not due to temporal changes in the age, sex, or major comorbidity burden of admitted patients.

## Acknowledgments

The authors wish to thank Alberto Sper Sempértegui, Diana C. Guanotoa Muñoz, Carlos P. Pérez Barona, Brayan A. Flores Reyes, Paul X. Garcés Villegas, Eliana M. Morejon Rosero,

Josué E. Castro Veintimilla, Rolando J. Chiluisa, Juan C. Jacome Guerrero, Wilson O. Echeverría Mora, Tatiana del Rocío Moreno Paz, Nery M. Cabrera Muñoz, Gerardo D. Zhunio Zhunio, Orlando A. Del Campo Torres, Maria I. Guanga Cadme, Ivonne Z. Peña Escalona, Zulay J. Ochoa Martinez who were very keen in filling the database of each patient recruited in the study.

## Author Contributions

**Conceptualization:** Pablo R. Morocho Jaramillo, Javier V. Velastegui Silva, Ana B. Ventimilla Campoverde.

**Data curation:** Katherine Simbaña-Rivera, Javier V. Velastegui Silva, Lenin Gómez-Barreno, Ana B. Ventimilla Campoverde, Juan F. Novillo Cevallos, Washington E. Almache Guanoquiza, Silvio L. Cedeño Guevara, Luis G. Imba Castro, Nelson A. Moran Puerta.

**Formal analysis:** Katherine Simbaña-Rivera, Pablo R. Morocho Jaramillo, Javier V. Velastegui Silva, Lenin Gómez-Barreno, Luis G. Imba Castro, Nelson A. Moran Puerta, Alex W. Guayta Valladares, Esteban Ortiz-Prado.

**Investigation:** Pablo R. Morocho Jaramillo, Juan F. Novillo Cevallos, Silvio L. Cedeño Guevara, Nelson A. Moran Puerta, Alex W. Guayta Valladares.

**Methodology:** Katherine Simbaña-Rivera, Lenin Gómez-Barreno, Alex W. Guayta Valladares, Esteban Ortiz-Prado.

**Resources:** Ana B. Ventimilla Campoverde, Juan F. Novillo Cevallos, Washington E. Almache Guanoquiza.

**Supervision:** Ana B. Ventimilla Campoverde, Silvio L. Cedeño Guevara, Luis G. Imba Castro.

**Validation:** Washington E. Almache Guanoquiza, Luis G. Imba Castro, Esteban Ortiz-Prado.

**Visualization:** Katherine Simbaña-Rivera.

**Writing – review & editing:** Alex Lister, Esteban Ortiz-Prado.

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
