## [Decision Letter · Decision Letter 0]

25 Aug 2021

PONE-D-21-07862

High-altitude is associated with better short-term survival in critically ill COVID-19 patients admitted to the ICU

PLOS ONE

Dear Dr. Ortiz-Prado,

Thank you for submitting your manuscript to PLOS ONE. After careful consideration, we feel that it has merit but does not fully meet PLOS ONE’s publication criteria as it currently stands. Therefore, we invite you to submit a revised version of the manuscript that addresses the points raised during the review process.

Both reviewers have identified aspects of your methods that require clarification, and pointed to deficiencies in the presentation of your manuscript that need to be rectified. Please pay careful attention to each of the concerns raised by the reviewers when preparing your revisions.

We look forward to receiving your revised manuscript.

Kind regards,

Jamie Males

Staff Editor

PLOS ONE

Journal Requirements:

3. Please include a caption for figures 2,3,4 and 5.

Reviewers' comments:

Reviewer's Responses to Questions

**Comments to the Author**

1. Is the manuscript technically sound, and do the data support the conclusions?

Reviewer #1: No

Reviewer #2: Partly

2. Has the statistical analysis been performed appropriately and rigorously? 

Reviewer #1: No

Reviewer #2: I Don't Know

3. Have the authors made all data underlying the findings in their manuscript fully available?

Reviewer #1: Yes

Reviewer #2: No

4. Is the manuscript presented in an intelligible fashion and written in standard English?

Reviewer #1: No

Reviewer #2: Yes

5. Review Comments to the Author

Reviewer #1: The hypothesis that high altitude (HA) protects against SARS-CoV-2 infection and COVID-19 has gained traction over the past several months, though admittedly not without contention. The hypothesis is supported both by epidemiological findings as well as biological plausibility for enhanced hypoxia tolerance in HA residents. Here, the authors tested whether short-term survival and ICU outcomes were different in HA versus low altitude regions. The question is undoubtedly significant and timely, though I have suggestions for improving data presentation and discussion.

Abstract: The background in the abstract is so vague “better or worse”, “lower or higher” that it’s confusing to follow what the authors hypothesize. Can APACHE II be defined? “Especially” higher is not quantitative. High altitude vs. high-altitude are used interchangeably. Please be consistent.

Introduction: the reference to HIF-1 as a protein that regulates angiogenesis is odd given all the other cytoprotective properties regulated by HIF that relate to COVID-19 pathogenesis.

Methods:

How do the two settings for HA and sea level differ? Are the cities of comparable size/rurality? Do they differ in quantitative metrics other than elevation?

The APACHE II and Charlson index are not described

What does “strict protocol implemented in both sites” (in the Bias section)?

Results and Discussion:

Given some of the demographic differences between HA and sea level, i wonder if the authors could statistically control for these factors in their analysis? If statistical regression were performed instead of Mann Whitney analyses, we could have more certainty that the differences in survival and lab biochemistry was really due to altitude rather than age and comorbidities, both of which are very different between elevations.

The hematological and serological parameters in Table 3 is simply too bulky to read. Can the authors present a portion of these outcomes which help directly test their hypothesis (and leave the others as supplemental data?). Same comment for the ventilation parameters

How standard were ventilation protocols between hospitals? For a non-expert, this data is really hard to understand.

Can repeated measures analysis be performed to see how some of these hematological and biochemical variables change over time, within the same patient at HA versus sea level? i.e. do HA patients return to normal faster than sea level?

The gasometry is interesting given that residence at HA should make these values different even at baseline. Is there a way to adjust for these, given that HA residence already impacts them? i.e. perhaps a relative change? The idea that SpO2/FiO2 serves as a prognostic factor is really underdeveloped

The txt states that tracheostomy was reached in 17.5% of patients, but the table says 18.7%.

The results for the hazard ratios is difficult to understand. Figure 1 should have five predictors of survival, yet it lists 6 (including PaO2/FiO2 twice, with two different numbers).

Figure 1 is used several times, and in general the figures are not organized correctly.

What is the difference between the survival curves and how were they generated?

The limitations section is simply too underdeveloped.

The references do not appear to be appropriately selected. For example, 23 and 24 should reference ACE2 polymorphisms and hypoxia tolerance, yet these references are reviews of ACE2 and ARB

Reviewer #2: Morocho Jaramillo and colleagues study potential differences of survival after severe Covid-19 infection related to the altitude of treatment. They find an increased survival time at higher altitude.

Generally, the manuscript is well written and the findings are interesting. However, in its current state it is very difficult to read for several reasons (many typos, figure numbers are incorrect, p-values in table seem to be incorrect) and appears not finalized.

Major:

1. Even though the median age of Covid19 patients was similar, the age distribution was clearly not. Much more individuals were in the highest age-categories (>56 years) in the low altitude group. This has to be considered when discussing the elevated death rate in the low altitude group. Without a way of showing, that this difference of age distribution was not the cause for the observed death rate differences, no influence of altitude can be assumed.

2. The figures are currently difficult to interpret:

a. The figure numbers are incorrect. All figure legends say figure 1. Figure 5(?) is the only figure located in the text and also in the end.

b. Please indicate for all figures (e.g. in the legends), which statistical measures were used (mean –median, SD-SEM-IQR …)

c. Fig 4(?): please check if legend is correct: panel C is indicated to be significantly different, but A and B not – seems not to be right

1. Please check all tables for accuracy; even though difficult to assess from IQRs some p-values are for example surprising in table 4 (esp. the many 0.000) and table 5 (e.g. heparins, p=0.42). Table 1: the p-values for the age categories are mostly 0.024, although sometimes (35-45) the differences do not appear significant. Also all the p=0.096 don’t seem right.

Minor:

1. page and line numbers would facilitate the review.

2. P10, Population and study size “We used a non-probabilistic sampling technique…” � this is formulated very vaguely; please be more specific on this sampling technique: was everybody included if the inclusion/exclusion criteria were fulfilled?

3. P11, Data sources and variables: “The data were obtained from the electronic medical record of a common registry system for both units and analyzed by three independent researchers.” � Please clarify what was analyzed independently and how the final results were obtained from these independent analyses.

4. P12, Bias: “… and a strict protocol was implemented in both sites.” � please specify

5. Results, Sociodemographic characteristic: “The BMI of all the patients was 27.8 kg/m2…” – please clarify that this is the median.

6. Discussion: “high-altitude patients present a chronic conversion of the hypoxia-inducing factor type 1 to type 2” – I am not sure, what this statement means, please clarify and add reference

7. What does it mean that the comorbidity rate was higher in lowlanders? May this reflect a higher incidence of those diseases at high altitude or could it also mean that a lower co-morbidity status may be associated with ICU admittance at higher altitude? Possibly indicating a higher risk to develop severe Covid19 at high altitude (even in absence of comorbidities? Or is it merely due to the unequal age-distribution of the 2 groups (see major point)? It would be important to look at this.

8. for the ACE-2 part: please consider the discussion of this topic in the cited work of Pun et al. Lower Incidence of COVID-19 at High Altitude: Facts and Confounders. High Altitude Medicine & Biology 2020

Figures and tables:

1. sometimes the variation seems to be indicated only on one side, while usually both sides are shown (e.g. Fig 2 (?), SaO2)

2. Table 1: CVA should be CVD?

3. Table 4: what is the difference btw. The 2 lines “resistors”?

Typos

Currently there are many typos in the manuscript, a non-exhaustive list is given below:

1. P10, The link between chronic exposure due to high altitude living and the clinical features of COVID-19 patients has been poorly studied. � remove “due”

2. P10 “…exposure on severally ill COVID-19 patients”… � should be “severely”?

3. in Fig 4(?) and Fig 5(?) legends :“comorbidtiesties”, “predictorsCaracterística”, “cuagolopathy” – also in Fig

4. incomplete sentences e.g. in the discussion: “In this way, it is evident that the strategy The ventilator used …”

5. …

Suggestions for further discussion points:

1. Could you discuss the result that the median BMI in both altitude categories is clearly above normal weight?

2. Is it possible to provide numbers of how many individuals per population were admitted to ICUs due to Covid19 in the investigated regions? This would not only be informative to put the present study in context but maybe also to understand some characteristics of the groups better – for example, why there is now difference in BMI between the groups, although lower BMIs are sometimes reported at higher altitudes.

3. Related to the previous point; are the catchment areas of the hospitals at low and high altitude comparable in terms of urban – rural, socioeconomic status, etc.?

4. Fig 2(?): Respiratory and physiological parameters among COVID-19 patients living at two different elevations: The great difference in SaO2 after 1 day should be discussed.

6. PLOS authors have the option to publish the peer review history of their article (what does this mean?). If published, this will include your full peer review and any attached files.

Reviewer #1: No

Reviewer #2: No

---

## [Author Response · Author response to Decision Letter 0]

22 Oct 2021

Point by Point Letter

To: 

Jamie Males

Staff Editor

PLOS ONE

RE: PONE-D-21-07862 High-altitude is associated with better short-term survival in critically ill COVID-19 patients admitted to the ICU

Dear Editor and reviewers, thank you very much for your observations and comments regarding our manuscript. Your observations and suggestions have improved our manuscript importantly. 

Please find enclosed our point-by-point response letter to each of your remarks.

We have updated the data availability statement and all the relevant data can be found here:

All non-identifiable and previously anonymized data can be retrieved from the following link to our open data digital repository: https://github.com/covid19ec/HospitalData. Any additional query or information about our research work can be requested to our email address at e.ortizprado@gmail.com

3. Please include a caption for figures 2,3,4 and 5.

We have added the missing captions within each new figure

Reviewers' comments:

Reviewer's Responses to Questions

Comments to the Author

1. Is the manuscript technically sound, and do the data support the conclusions?

Reviewer #1: No

Reviewer #2: Partly

We have improved our manuscript significantly; we hope this second revision will fulfil your expectations 

2. Has the statistical analysis been performed appropriately and rigorously? 

Reviewer #1: No

Reviewer #2: I Don't Know

We have improved our statistical analyses, incorporating all your suggestions 

3. Have the authors made all data underlying the findings in their manuscript fully available?

Reviewer #1: Yes

Reviewer #2: No

We have updated the data availability statement and all the relevant data can be found here:

4. Is the manuscript presented in an intelligible fashion and written in standard English?

Reviewer #1: No

Reviewer #2: Yes

We have incorporated your comments, which has significantly improved the quality of our work

5. Review Comments to the Author

Reviewer #1: The hypothesis that high altitude (HA) protects against SARS-CoV-2 infection and COVID-19 has gained traction over the past several months, though admittedly not without contention. The hypothesis is supported both by epidemiological findings as well as biological plausibility for enhanced hypoxia tolerance in HA residents. Here, the authors tested whether short-term survival and ICU outcomes were different in HA versus low altitude regions. The question is undoubtedly significant and timely, though I have suggestions for improving data presentation and discussion.

Thanks for your comments, we have reviewed the manuscript entirely and added all your observations 

Abstract: The background in the abstract is so vague “better or worse”, “lower or higher” that it’s confusing to follow what the authors hypothesize. Can APACHE II be defined? “Especially” higher is not quantitative. High altitude vs. high-altitude are used interchangeably. Please be consistent.

Thanks for your observations, we have improved the manuscript as follow:

Background: Multiple studies have attempted to elucidate the relationship between chronic hypoxia and SARS-CoV-2 infection. It seems that high altitude is associated with lower COVID-19 related mortality and incidence rates, nevertheless, all the data came from observational studies, being this the first one looking into prospectively-collected clinical data from severely ill patients residing at two significantly different altitudes. 

Methods: A prospective cohort, two-center study among COVID-19 confirmed adult patients admitted to a low (sea level) and high altitude (2,850 m) ICU unit in Ecuador was conducted. Two hundred and thirty confirmed patients were enrolled from March 15th to July 15th, 2020. 

Results: From 230 patients, 149 were men (64.8%) and 81 women (35.2%). The median age of all the patients was 60 years and at least 105 (45.7%) of patients had at least one underlying comorbidity including hypertension (33.5%), diabetes (16.5%), and chronic kidney failure (5.7%). The APACHE II scale (Score that estimates ICU mortality) at 72 hours was especially higher in the low altitude group with a median of 18 points (IQR: 9.5-24.0), compared to 9 points (IQR: 5.0-22.0) obtained in the group of high altitude. There is evidence of a difference in survival in favor of the high altitude group (p = 0.006), the median survival being 39 days, compared to 21 days in the low altitude group. 

Conclusion: There has been a substantial improvement in survival amongst people admitted to the high altitude ICU. Residing at High altitude was associated with improved survival, especially among patients with no comorbidities. COVID-19 patients admitted to the high altitude ICU unit have improved severity-of-disease classification system scores at 72 hours.

Introduction: the reference to HIF-1 as a protein that regulates angiogenesis is odd given all the other cytoprotective properties regulated by HIF that relate to COVID-19 pathogenesis.

Thanks so much for your observation, we have updated our literature review on this very topic as follow:

During the first months of the pandemic, very few ecological studies showed a possible epidemiological and survival implication exerted by high altitude7,8. It has been proposed that these results are in part answered by the well-known physiological acclimatization and the long term adaptation to high altitude exposure, generating a greater tolerance to chronic hypoxia 7–10. Several investigations have tried to determine the potential relationship between high altitude and COVID-19 related mortality 1–4. Most studies have established that living at high altitudes could be related to reduced COVID-19 related mortality and morbidity 1–3 . These results were discussed from different points of view. The biological one was based on a hypothesized lower viral affinity for the type-2 angiotensin-converting enzyme (ACE2) receptors, but there is no definitive evidence to supporting this claim 1. Another hypothesis surrounding the high altitude-COVID-19 link refers to the involvement of better perfused and better oxygenated tissues due to the involvement of the hypoxia-triggered protein that regulates angiogenesis, cell proliferation, metabolism and downregulates ACE-2 levels; the well-known Hypoxia-inducible factor 1-alpha (HIF-1 α) 1–7. Having improved ability to use oxygen within the tissues might reduce the effects of systemic hypoxia caused by acute respiratory distress syndrome (ARDS)1.

On the other hand, sociodemographic and environmental factors such as population density, UV radiation, ozone or cold have been proposed to affect SARS-CoV-2 transmission and viral load, nevertheless, no clinical data is available yet 1,2. 

The link between high altitude hypoxia and COVID-19 mortality is still under investigation17,18. The very few studies on clinical, ventilatory and respiratory support parameters’ differences have been performed at elevation below 1,500 m and no evidence about the role of high altitude exposure (> 2,500 m) on severally ill COVID-19 patients have been published yet 19.

We believe our study is the first one that has been able to demonstrate the effect of altitude living on COVID-19 mortality and prognosis after controlling for several clinical factors. 

Methods:

How do the two settings for HA and sea level differ? Are the cities of comparable size/rurality? Do they differ in quantitative metrics other than elevation?

We have added the following paragraph within the settings section of the manuscript

Quito and Guayaquil are the most important cities in Ecuador with 2.5 and 2.9 million inhabitants respectively. The two ICU were built on 2017, both are part of the same Social Security Health System model (IESS) and both were the first COVID-19 sentinel hospitals in Ecuador. Since both hospitals were the first in receiving patients during the pandemic, both used the same therapeutical protocols and receive the same type of patients. 

The APACHE II and Charlson index are not described

Thanks for your comments, we have added the meaning of both scales. 

What does “strict protocol implemented in both sites” (in the Bias section)?

We have updated our bias section and included other measures used to avoid bias.

Results and Discussion:

Given some of the demographic differences between HA and sea level, i wonder if the authors could statistically control for these factors in their analysis? If statistical regression were performed instead of Mann Whitney analyses, we could have more certainty that the differences in survival and lab biochemistry was really due to altitude rather than age and comorbidities, both of which are very different between elevations.

Thanks so much for your suggestion, we have performed a subgroup analysis and is included within the results section.

The hematological and serological parameters in Table 3 is simply too bulky to read. Can the authors present a portion of these outcomes which help directly test their hypothesis (and leave the others as supplemental data?). Same comment for the ventilation parameters

Thanks so much for your suggestion. We have reduced the number of variables within the tables

How standard were ventilation protocols between hospitals? For a non-expert, this data is really hard to understand.

A paragraph has been added at the beginning of the results clearly explaining the differences between the ventilatory parameters used.

Can repeated measures analysis be performed to see how some of these hematological and biochemical variables change over time, within the same patient at HA versus sea level? i.e. do HA patients return to normal faster than sea level?

Unfortunately, we only have 3 temporary measurements that do not allow us to perform the type of analysis requested. 

The gasometry is interesting given that residence at HA should make these values different even at baseline. Is there a way to adjust for these, given that HA residence already impacts them? i.e. perhaps a relative change? The idea that SpO2/FiO2 serves as a prognostic factor is really underdeveloped.

Thank you for your comments, we were not able to recreate a model that could fit the gasometric values reported at high altitude with other variables, however we have added your observation to the limitations section

The txt states that tracheostomy was reached in 17.5% of patients, but the table says 18.7%.

Thanks for pointing this out, we have corrected the mistake. 

The results for the hazard ratios is difficult to understand. Figure 5 should have five predictors of survival, yet it lists 6 (including PaO2/FiO2 twice, with two different numbers).

Figure 5 has been modified to ensure a better understanding of the results. 

Figure 1 is used several times, and in general the figures are not organized correctly.

We have updated all the figures and tables

What is the difference between the survival curves and how were they generated?

“Bivariate and multivariate analyzes were performed to identify factors associated with death from COVID-19 in all patients using the Cox risk regression model. To obtain a reduced set of variables from the broad set of predictors, we carried out a progressive in bloc procedure assigning the predictor variables into six groups: sociodemographic characteristics and comorbidities, complications, scales, ventilatory values, medications, and laboratory and imaging parameters. A multivariate regression analysis was applied within each block using two criteria to achieve the best set of predictors: relevance to the clinical situation and bivariate and multivariate statistical significance (p <0.05). Variables with more than 25% missing values were not considered for the analysis.”

The limitations section is simply too underdeveloped.

We have updated this section according to all the comments generated by the reviewers 

The references do not appear to be appropriately selected. For example, 23 and 24 should reference ACE2 polymorphisms and hypoxia tolerance, yet these references are reviews of ACE2 and ARB

We have updated the entire references list 

Reviewer #2: Morocho Jaramillo and colleagues study potential differences of survival after severe Covid-19 infection related to the altitude of treatment. They find an increased survival time at higher altitude.

Generally, the manuscript is well written and the findings are interesting. However, in its current state it is very difficult to read for several reasons (many typos, figure numbers are incorrect, p-values in table seem to be incorrect) and appears not finalized.

Thanks for your observations, we have updated the entire manuscript 

Major:

1- Even though the median age of Covid19 patients was similar, the age distribution was clearly not. Much more individuals were in the highest age-categories (>56 years) in the low altitude group. This has to be considered when discussing the elevated death rate in the low altitude group. Without a way of showing, that this difference of age distribution was not the cause for the observed death rate differences, no influence of altitude can be assumed.

The results of the age subgroups were reviewed. Since they were not standardized with population distributions as recommended by the WHO and since the statistical test that established the association was a 6x2 chi test table, the authors decided to eliminate this confounding factor from the study and maintain the quantitative measure expressed in the nonparametric statistics. We have added this to the limitation section

2. The figures are currently difficult to interpret: 

a. The figure numbers are incorrect. All figure legends say figure 1. Figure 5(?) is the only figure located in the text and also in the end.

Thanks for your observation, we have updated all the figures and tables 

b. Please indicate for all figures (e.g. in the legends), which statistical measures were used (mean –median, SD-SEM-IQR …)

Thanks for your observation, we have updated all the figures and tables 

c. Fig 4(?): please check if legend is correct: panel C is indicated to be significantly different, but A and B not – seems not to be right

Thanks for your observation, we have updated all the figures and tables 

1. Please check all tables for accuracy; even though difficult to assess from IQRs some p-values are for example surprising in table 4 (esp. the many 0.000) and table 5 (e.g. heparins, p=0.42). Table 1: the p-values for the age categories are mostly 0.024, although sometimes (35-45) the differences do not appear significant. Also all the p=0.096 don’t seem right.

The error in the comparative calculation of age has been corrected, the content of the tables has been reduced and in Table 4, a detailed explanation of the data has been included, which, although they show statistical differences, these differences are not clinically relevant. 

Minor:

1. page and line numbers would facilitate the review. 

Many thanks for your observations, we have added continuous lines

2. P10, Population and study size “We used a non-probabilistic sampling technique…” � this is formulated very vaguely; please be more specific on this sampling technique: was everybody included if the inclusion/exclusion criteria were fulfilled? 

The wording has been improved and specified that all patients who met the inclusion and exclusion criteria

3. P11, Data sources and variables: “The data were obtained from the electronic medical record of a common registry system for both units and analyzed by three independent researchers.” � Please clarify what was analyzed independently and how the final results were obtained from these independent analyses.

The clinical data were obtained by the intensivists of both hospitals; however, the analysis was performed by 3 of the physicians who did not collect the data, limiting the role of research bias while collecting all the information 

4. P12, Bias: “… and a strict protocol was implemented in both sites.” � please specify

The bias section was improved thanks to your observations

5. Results, Sociodemographic characteristic: “The BMI of all the patients was 27.8 kg/m2…” – please clarify that this is the median.

We have updated this information 

6. Discussion: “high-altitude patients present a chronic conversion of the hypoxia-inducing factor type 1 to type 2” – I am not sure, what this statement means, please clarify and add reference

We have improved our discussion section, making it clearer and correctly referencing when needed. 

7. What does it mean that the comorbidity rate was higher in lowlanders? May this reflect a higher incidence of those diseases at high altitude or could it also mean that a lower co-morbidity status may be associated with ICU admittance at higher altitude? Possibly indicating a higher risk to develop severe Covid19 at high altitude (even in absence of comorbidities? Or is it merely due to the unequal age-distribution of the 2 groups (see major point)? It would be important to look at this.

Thanks so much for pointing this out, we have clarified and highlighted our response as follow:

Comorbidities are a well-associated with an increasing risk of COVID-19 related mortality1,2. We found that mortality is positively associated with having one or more comorbidities, however in our study we found that although the presence of comorbidities is higher in populations located at lower altitudes, once we excluded the presence of comorbidities from the model, the hypothesized protective effect of high altitude is evident. In other words, patients with comorbidities are at higher risk of dying at both altitudes when compared to patients with no comorbidities, nevertheless, when compared only patients without comorbidities from the low and high altitude group, we found that highlanders have greater chance of survival. 

8. for the ACE-2 part: please consider the discussion of this topic in the cited work of Pun et al. Lower Incidence of COVID-19 at High Altitude: Facts and Confounders. High Altitude Medicine & Biology 2020

The reference was added and discussed as follow:

Other physiological mechanisms could justify, at least in part, this apparent protection conferred by geographical altitude. It is believed that at high altitude there is a lower expression of ACE-2 receptors, which are precisely the gateway to our cells for the SARS-CoV-2 virus1. A more plausible explanation goes along with the fact that high altitude inhabitants express genes responsible for producing more erythrocytes (increasing oxygen transport) and creating new blood vessels (greater oxygen supply)1–3. On top of this, we must add certain anatomical and morphological characteristics, among high altitude dwellers such as larger and bigger thoraxes as well as greater ventilatory capacities, that might play a role reducing hypoxia found during severe ARDS due to COVID-191–4.

Figures and tables:

1. sometimes the variation seems to be indicated only on one side, while usually both sides are shown (e.g. Fig 2 (?), SaO2)

All the figures and tables were updated 

2. Table 1: CVA should be CVD?

All the figures and tables were updated, CVD is the correct abbreviation 

3. Table 4: what is the difference btw. The 2 lines “resistors”?

We have eliminated this mistake and rephrase the paragraph 

Typos

Currently there are many typos in the manuscript, a non-exhaustive list is given below:

Many thanks for your time and observations, we have reviewed the entire manuscript for mistakes and typos

1. P10, The link between chronic exposure due to high altitude living and the clinical features of COVID-19 patients has been poorly studied. � remove “due”

Thanks, this was done

2. P10 “…exposure on severally ill COVID-19 patients”… � should be “severely”?

Thanks, this was corrected 

3. in Fig 4(?) and Fig 5(?) legends :“comorbidtiesties”, “predictorsCaracterística”, “cuagolopathy” – also in Fig

Thanks, this was reviewed and corrected 

4. incomplete sentences e.g. in the discussion: “In this way, it is evident that the strategy The ventilator used …”

We have completed the sentences, thanks 

Suggestions for further discussion points:

1. Could you discuss the result that the median BMI in both altitude categories is clearly above normal weight?

Yes, an entire paragraph was added as follow:

Our results demonstrate that the presence of overweight and obesity were consistent characteristics of both groups. Current evidence is clear linking obesity as an independent predictor of mortality among COVID-19 patients1. Our study has similar results to a large UK study, which confirmed that 44% of hospitalized patients were overweight and 34% obese1. The information suggests that after adjusting for possible confounding factors, including age, sex, ethnicity and social deprivation, the relative risk of critical illness from COVID-19 increases by 44% for overweight people and almost doubles for obese people.

2. Is it possible to provide numbers of how many individuals per population were admitted to ICUs due to Covid19 in the investigated regions? This would not only be informative to put the present study in context but maybe also to understand some characteristics of the groups better – for example, why there is now difference in BMI between the groups, although lower BMIs are sometimes reported at higher altitudes.

The present study included a total of 230 patients diagnosed with COVID-19 using the RT-PCR technique, of which 114 patients were treated in the high altitude group (IESS-Quito Sur), while 116 patients belonged to the low altitude group (IESS-Los Ceibos). 

3. Related to the previous point; are the catchment areas of the hospitals at low and high altitude comparable in terms of urban – rural, socioeconomic status, etc.?

We have updated this information was also requested by the other reviewer 

4. Fig 2(?): Respiratory and physiological parameters among COVID-19 patients living at two different elevations: The great difference in SaO2 after 1 day should be discussed.

 We have added an entire paragraph on this

---

## [Decision Letter · Decision Letter 1]

5 Nov 2021

PONE-D-21-07862R1High altitude is associated with better short-term survival in critically ill COVID-19 patients admitted to the ICUPLOS ONE

Dear Dr. Ortiz-Prado,

Thank you for submitting your manuscript to PLOS ONE. After careful consideration, we feel that it has merit but does not fully meet PLOS ONE’s publication criteria as it currently stands. Therefore, we invite you to submit a revised version of the manuscript that addresses the points raised during the review process.

We look forward to receiving your revised manuscript.

Kind regards,

Danielle R. Bruns, PhD

Academic Editor

PLOS ONE

Journal Requirements:

Additional Editor Comments:

Thank you for your revised manuscript. We appreciate the attention you took in addressing each reviewer's comments. I disclose that I participated as a reviewer for the initial evaluation of this manuscript. Upon second review, one reviewer still has a few concerns. While the manuscript is substantially improved, the major contribution of age in high- versus low-altitude communities is still inadequately addressed. Given the significant impact of age on COVID pathogenesis, we believe this is still a major concern in the revised version. The authors need to address this concern statistically (correct for the difference in age) or to at least discuss the importance of age in COVID-19. Secondly, some concerns remain with respect to the discussion of HIF-1, especially with the reference provided. Minor comments also remain regarding figure numbering.

Reviewers' comments:

Reviewer's Responses to Questions

**Comments to the Author**

1. If the authors have adequately addressed your comments raised in a previous round of review and you feel that this manuscript is now acceptable for publication, you may indicate that here to bypass the “Comments to the Author” section, enter your conflict of interest statement in the “Confidential to Editor” section, and submit your "Accept" recommendation.

Reviewer #2: (No Response)

2. Is the manuscript technically sound, and do the data support the conclusions?

Reviewer #2: Partly

3. Has the statistical analysis been performed appropriately and rigorously? 

Reviewer #2: Yes

4. Have the authors made all data underlying the findings in their manuscript fully available?

Reviewer #2: Yes

5. Is the manuscript presented in an intelligible fashion and written in standard English?

Reviewer #2: Yes

6. Review Comments to the Author

Reviewer #2: The authors have substantially improved the manuscript and most of my comments have been adequately addressed. However, 2 important concerns (+1 formal point) remain for me:

1. the difference in age distribution (much more individuals were in the highest age-categories (>56 years) in the low altitude group): although the problem is now mentioned in the limitations, this difference in distribution could explain the apparent altitude effect, if it were mostly the older people (>56) that died. I think it would be very important to correct for the age effect – at least the importance of age on mortality from COVID-19 has to be discussed. Currently the text reads, like age is a negligible factor in COVID-19 risk, which it clearly is not.

2. the argumentation about HIF-1 in high altitude dwellers. Unfortunately, the provided reference is behind a paywall (and most likely is not the ideal reference for this purpose), so it might be a confusion on my side: but the following statement needs clarification:

“High altitude patients present a chronic molecular conversion of the hypoxia-inducing factor type 1 to type 2 (HIF-1), which favors a greater tolerance to hypoxemia and decreases the acute tissue damage triggered by patients with severe acute respiratory conditions41”

I am not sure, what is meant with “conversion of HIF-1 type 1 to type 2”. Does this simply refer to increased stabilization of HIF-1alpha in high altitude residents? Does it refer to the interaction of HIF1- and HIF-2 pathways? Or to genetic adaptations in HIF-2alpha associated with its enhanced activity in some high altitude populations (such as in Tibetans)? If any of these effects are eluded to, the sentence needs to be more precise (a review that would provide adequate references is for example this one: Bigham & Lee. Human high-altitude adaptation: forward genetics meets the HIF pathway. Genes Dev. 2014). Otherwise, I think better explanation has to be provided for what is meant with “type 1 and type 2”.

3. In addition, please note, that all figures are still labeled Fig. 1.

7. PLOS authors have the option to publish the peer review history of their article (what does this mean?). If published, this will include your full peer review and any attached files.

Reviewer #2: **Yes: **Johannes Burtscher

---

## [Author Response · Author response to Decision Letter 1]

19 Dec 2021

Point by Point Letter

To: 

Danielle R. Bruns, PhD

Academic Editor

PLOS ONE

RE: PONE-D-21-07862R1 “High-altitude is associated with better short-term survival in critically ill COVID-19 patients admitted to the ICU”

Dear Editor and reviewers, thank you very much for your effort, observing our manuscript for the second time. We have completed the revision and we have fulfilled your comments. All the changes are highlighted in red. We have also included a clean version and the rebuttal letter.

Additional Editor Comments:

Thank you for your revised manuscript. We appreciate the attention you took in addressing each reviewer's comments. I disclose that I participated as a reviewer for the initial evaluation of this manuscript. Upon second review, one reviewer still has a few concerns. While the manuscript is substantially improved, the major contribution of age in high- versus low-altitude communities is still inadequately addressed. Given the significant impact of age on COVID pathogenesis, we believe this is still a major concern in the revised version. The authors need to address this concern statistically (correct for the difference in age) or to at least discuss the importance of age in COVID-19. 

Dear Editor, we have responded to your observations, and we believe that the further analysis improved our manuscript. 

Secondly, some concerns remain with respect to the discussion of HIF-1, especially with the reference provided. 

We have addressed this section and included a deeper analysis within the discussion section

Minor comments also remain regarding figure numbering.

We have reviewed the entire manuscript and corrected all the typos and mistakes while numbering tables and figures.

6. Review Comments to the Author

Reviewer #2: The authors have substantially improved the manuscript and most of my comments have been adequately addressed. However, 2 important concerns (+1 formal point) remain for me:

Thanks for your comments, we have improved our second revision

1. the difference in age distribution (much more individuals were in the highest age-categories (>56 years) in the low altitude group): although the problem is now mentioned in the limitations, this difference in distribution could explain the apparent altitude effect, if it were mostly the older people (>56) that died. I think it would be very important to correct for the age effect – at least the importance of age on mortality from COVID-19 has to be discussed. Currently the text reads, like age is a negligible factor in COVID-19 risk, which it clearly is not.

Thanks for your keen observation, we agreed with you, and we have corrected our findings after adjusting our results for age differences. The new analysis are highlighted in red and new surviving curves were elaborated 

2. the argumentation about HIF-1 in high altitude dwellers. Unfortunately, the provided reference is behind a paywall (and most likely is not the ideal reference for this purpose), so it might be a confusion on my side: but the following statement needs clarification:

“High altitude patients present a chronic molecular conversion of the hypoxia-inducing factor type 1 to type 2 (HIF-1), which favors a greater tolerance to hypoxemia and decreases the acute tissue damage triggered by patients with severe acute respiratory conditions41”

I am not sure, what is meant with “conversion of HIF-1 type 1 to type 2”. Does this simply refer to increased stabilization of HIF-1alpha in high altitude residents? Does it refer to the interaction of HIF1- and HIF-2 pathways? Or to genetic adaptations in HIF-2alpha associated with its enhanced activity in some high altitude populations (such as in Tibetans)? If any of these effects are eluded to, the sentence needs to be more precise (a review that would provide adequate references is for example this one: Bigham & Lee. Human high-altitude adaptation: forward genetics meets the HIF pathway. Genes Dev. 2014). Otherwise, I think better explanation has to be provided for what is meant with “type 1 and type 2”.

Many thanks for your comments, we have deepened our discussion on this very subject and clarified the doubt raised by you.

3. In addition, please note, that all figures are still labeled Fig. 1.

Thanks for this observation, we have numbered them accordingly.

---

## [Editor Report · Decision Letter 2]

27 Dec 2021

High altitude is associated with better short-term survival in critically ill COVID-19 patients admitted to the ICU

PONE-D-21-07862R2

Dear Dr. Ortiz-Prado,

We’re pleased to inform you that your manuscript has been judged scientifically suitable for publication and will be formally accepted for publication once it meets all outstanding technical requirements.

Kind regards,

Danielle R. Bruns, PhD

Guest Editor

PLOS ONE

Additional Editor Comments (optional):

We thank the authors for responding to all reviewer and editorial comments.
---

## [Editor Report · Acceptance letter]

22 Mar 2022

PONE-D-21-07862R2 

High-altitude is associated with better short-term survival in critically ill COVID-19 patients admitted to the ICU 

Dear Dr. Ortiz-Prado:

I'm pleased to inform you that your manuscript has been deemed suitable for publication in PLOS ONE. Congratulations! Your manuscript is now with our production department. 

Kind regards, 

on behalf of

Dr. Danielle R. Bruns 

Guest Editor

PLOS ONE